# Wheelchair Badminton: A Narrative Review of Its Specificities

Ilona Alberca [1,*], Bruno Watier [2,3], Félix Chénier [4,5], Florian Brassart [1] and Arnaud Faupin [1]

[1] Laboratoire J-AP2S, Université de Toulon, 83130 La Garde, France; florian.brassart@outlook.fr (F.B.); faupin@univ-tln.fr (A.F.)

[2] Laboratoire d'Analyse et d'Architecture des Systèmes-Centre National de la Recherche Scientifique (LAAS-CNRS), Université de Toulouse, CNRS, Université Toulouse III Paul Sabatier (UPS), 31058 Toulouse, France; bruno.watier@laas.fr

[3] Centre National de la Recherche Scientifique-Advanced Industrial Science and Technology (CNRS-AIST JRL (Joint Robotics Laboratory)), International Research Laboratories (IRL), National Institute of Advanced Industrial Science and Technology (AIST), 1-1-1 Umezono, Tsukuba 305-8560, Ibaraki, Japan

[4] Mobility and Adaptive Sports Research Lab, Department of Physical Activity Sciences, Université du Québec à Montréal, Montreal, QC H2X 3J8, Canada; chenier.felix@uqam.ca

[5] Centre for Interdisciplinary Research in Rehabilitation of Greater Montreal, Institut Universitaire sur la Réadaptation en Déficience Physique de Montréal, Montreal, QC H3S 2J4, Canada

* Correspondence: ilona.alberca@univ-tln.fr; Tel.: +33-7-86-49-09-57

**Abstract:** This narrative review aims to provide a comprehensive overview of the scientific literature on wheelchair badminton and its distinctive aspects, encompassing abilities and disabilities, propulsion technique, and the use of a racket. The relatively young history of wheelchair badminton is reflected in the scarcity of scientific studies within this domain, highlighting the need for further investigation. The review systematically covers existing articles on wheelchair badminton, offering a nearly exhaustive compilation of knowledge in this area. Findings suggest that athletes with abdominal capacities engage in more intense matches with a higher frequency of offensive shots compared to athletes with little or no abdominal capacities. Moreover, backward propulsion induces higher cardiorespiratory responses, overall intensity of effort, physiological stress, metabolic load, and rating of perceived exertion, particularly at high imposed rolling resistance or speeds, and makes it difficult to generate sufficient forces on the handrim, requiring adjustments in the kinematics of propulsion techniques, particularly at high rolling resistances or speeds, potentially leading to performance decrements. The use of a badminton racket further increases generated forces while decreasing the efficiency of propulsion and modifying the propulsion technique with shorter and quicker pushes, potentially impacting performance. Further research is imperative to explore additional perspectives, address existing gaps, and expand the scope of study within the wheelchair badminton domain. This narrative review serves as a foundation for future investigations, emphasizing the necessity of continued research to enhance our understanding of wheelchair badminton.

**Keywords:** biomechanics; badminton particularities; racket; classification; propulsion direction



## 1. Introduction

Badminton is a globally popular sport with recognized physical and mental health benefits [1]. However, physical disabilities often hinder individuals from engaging in this activity. Wheelchair badminton emerged as an adapted form of the sport, facilitating the participation of individuals with physical disabilities using a wheelchair in their daily lives, even in competitive settings, thereby allowing them to experience the advantages of badminton [2]. Originating in the 1990s, wheelchair badminton started when several German athletes became interested in adapting the rules of able-bodied badminton to allow for the inclusion of individuals with disabilities [3]. Wheelchair badminton follows the rules established and unified worldwide by the BWF and is similar to able-bodied badminton. The reduced number of athletes enables gender and disability inclusivity [3].

Despite being a relatively recent discipline, wheelchair badminton is taking its place on the international stage thanks to its inclusion in the Tokyo 2021 Paralympic Games.

Several specificities unique to wheelchair badminton exist. The classification system, specific to wheelchair badminton, includes the creation of two wheelchair classes with characteristics unique to each class and refers to the abilities and disabilities of each athlete. These athletes can be categorized into two classes: wheelchair 1 (WH1) and wheelchair 2 (WH2). The WH1 class corresponds to manual wheelchair users with abdominal and lower limb paralysis, while the WH2 class corresponds to users with abdominal capabilities but lower limb paralysis with partial sensation and who may sometimes move in a vertical position using crutches or prostheses but only practice the sport in a wheelchair [3–5]. The athlete's classification process involves determining eligibility based on their level of impairment as described in BWF [5] and then assigning the athlete to his or her class based on a technical and physical assessment [5].

After verifying the athlete's eligibility, he will undergo a physical and technical evaluation taking place during a tournament or training match. Assessors are asked to identify the following profiles:

- WH1: "Players demonstrate functional limitation based on muscle power and range of motion of the trunk and possibly upper limbs during a match or training session." (BWF);
- WH2: "Players have a functional limitation based on limited muscle power or range of motion, requiring the use of walking aids. A shift in the center of gravity may result in loss of balance, for example when attempting to pivot or stop and start." [5].

Depending on the person's eligible disability type, these profiles are refined to determine specific impairments and classify athletes as WH1 or WH2. Athletes are also subjected to movements specific to this sport: alternating forward and backward propulsion with little or no lateral movement. Finally, the equipment used in wheelchair badminton practice does not differ from regular badminton, and athletes have to propel themselves with the use of a racket. We know that the use of a racket can have an impact on an athlete's performance, as shown by studies carried out in the closely related discipline of wheelchair tennis [6–8]. Indeed, these studies revealed the following aspects when using the racket:

- Reduced velocity of athletes [7];
- Negative effects on the propulsion technique and may lead to injuries of the upper extremity due to the longer time needed to couple the hand with the racket to the rim [6];
- Reduction in propulsive moment may lead to a decrease in performance [8].

Overall, these results may indicate a reduction in the performance of athletes due to changes in their kinetics, kinematics, and spatiotemporal parameters when using a racket [6–8]. In light of these specificities, the inclusion of biomechanical analysis would enhance our understanding of wheelchair badminton, particularly concerning performance. Thus, this narrative review aims to provide a biomechanical overview of the literature on wheelchair badminton and its specificities: the athletes' abilities related to their classification, the direction of propulsion, and the use of a racket. The goal is to assess what has been undertaken in the literature so far, draw potential recommendations for athletes, and guide future research in this field.

## 2. Materials and Methods

The present narrative review is a systematic examination of the various specific features of wheelchair badminton: the sport's specific classification, the use of a racket to propel the wheelchair, and the predominantly forward and backward propulsion movements. This review provides an overview of current knowledge to guide future research in this field. The aim was to focus exclusively on wheelchair badminton players who use a wheelchair in their daily lives. Given the novelty of this sport, the paucity of scientific studies available, and based on the narrative review by Bakatchina et al. [9], a narrative review was considered to be methodologically more appropriate than a systematic review. To

conduct this narrative review, we used the following three research algorithms to address the three specific features of wheelchair badminton:

- "wheelchair" AND "para badminton" OR "para-badminton" OR "parabadminton" AND "classification" AND "WH1" OR "WH2";
- "wheelchair" AND "para badminton" OR "para-badminton" OR "parabadminton" AND "classification" AND "racket";
- "wheelchair" AND "para badminton" OR "para-badminton" OR "parabadminton" AND "classification" AND "forward propulsion" OR "backward propulsion" OR "reverse propulsion".

Those algorithms were applied to the PubMed and Cairn databases. Google Scholar was also employed to verify the research conducted on the two databases. The articles were selected based on titles and abstracts and, subsequently, considering the entirety of the text. The articles were included if they were in English or French, focused on a population of wheelchair badminton athletes with a motor disability, and addressed biomechanical data analysis. Considering the limited number of articles obtained from these searches, it was decided to include articles on able-bodied populations and on technical, temporal, and physiological data to be as comprehensive/exhaustive as possible regarding the three specific aspects of badminton. In total, two articles were included on the classification issue, three on the direction of propulsion, and two on the impact of the badminton racket. Table 1 summarizes the various included studies.

**Table 1.** Summary of the different studies included in this narrative review.

| Study | Type | Aim | Participants | Wheelchair | Design | Measurments Tools | Parameters |
|---|---|---|---|---|---|---|---|
| Strapasson (2021) [10] | Research article | Investigate technical and temporal parameters in WH1 and WH2 | 42 international wheelchair badminton players (33 WH1; 25 WH2) | Participants' own wheelchairs | Video analysis of 20 men's singles matches during the 11th World PBd Championship | 3 camcorders | Technical and temporal wheelchair badminton match parameters |
| Mota Ribeiro & de Almeida (2020) [11] | Research article | Describe and compare the temporal and technical characteristics | Brazilian international wheelchair badminton players | Participants' own wheelchairs | Video analysis of 23 men's singles matches during the 2018 Brazilian Para-badminton Championship | 4 GoPro Hero Silver cameras, Windows Media Player software, digital timer | Technical and temporal wheelchair badminton match parameters |
| Linden et al. (1993) [12] | Research article | Compare metabolic and cardiopulmonary responses | 21 moderately active able-bodied males | Backless bench between two independently uncambered wheelchair wheels | Propel for 3 min at 4 different imposed power outputs (15, 20, 25, 30 W) | 2 cyclosimulators, indirect calorimetry, electronic revolution counter, Collins two-way breathing valve, Quinton Q-Plex I metabolic cart, electrocardiogram | Physiological and biomechanical parameters |
| Salvi et al. (1998) [13] | Research article | Compare the physiologic demands of forward and reverse wheeling techniques | 10 able-bodied men | Standard wheelchair with no wheel camber and no arm rests | Propel for 3 min at 6 different imposed power outputs (2.5, 6.0, 12.5, 18.5, 25.0, 30.0 W) | Ergometer, open circuit spirometry, calibrated electronic oxygen analyzer, carbon dioxide analyzers, calibrated dry gas meter, telemetry system, automated lactate analyzer | Physiological and biomechanical parameters |
| Mason et al. (2015) [14] | Research article | Explore physiological and biomechanical differences | 14 able-bodied males with previous wheelchair propulsion experience | Sports wheelchair with 15° rear-wheel camber | Propel for 3 min at 3 sub-maximal imposed speeds (4, 6, 8 km/h) | Single-roller ergometer, 1 instrumented wheel, breath-by-breath system, radio telemetry | Physiological and biomechanical parameters |
| Alberca et al. (2022) [15] | Research article | Investigate the impact of the badminton racket on kinetic and spatiotemporal parameters | 16 able-bodied sports students introduced to wheelchair badminton | Multi-sport wheelchair with a 26-inch wheel size and an 18° camber angle | Propel with and without a racket held on the dominant side along a 20m straight line at a constant velocity of 5 km/h | 2 instrumented wheels | Biomechanical parameters |

**Table 1.** *Cont.*

| Study | Type | Aim | Participants | Wheelchair | Design | Measurments Tools | Parameters |
|---|---|---|---|---|---|---|---|
| Fukui et al. (2020) [16] | Short communication | Investigate the difference in forearm muscle fatigue | 6 able-bodied men | Not specified | Maximal repetitions of 30-cm forward–back sprints using a wheelchair for 20 s under 2 conditions: with and without a racquet | Muscle hardness tester, near-infrared spectroscopy | Muscle parameters |

## 3. Abilities and Disabilities

Sports classification in the field of disability is crucial, and the aim is to equalize opportunities and thus create fairer competitions for everyone. Since a player's disability limits their ability to play a sport, this limitation must be recognized and therefore requires classification [17]. The wheelchair badminton classification was created by the BWF to meet the requirements of the International Paralympic Committee Athlete Classification Code 2015 and international standards. The system is highly inclusive since it allows people with various disabilities to play wheelchair badminton. Athletes with a motor impairment who use a wheelchair in their daily lives can be categorized into two classes: wheelchair 1 (WH1) and wheelchair 2 (WH2). The WH1 class corresponds to manual wheelchair users with abdominal and lower limb paralysis, while the WH2 class corresponds to users with abdominal capabilities but lower limb paralysis with partial sensation and who may sometimes move in a vertical position using crutches or prostheses but only practice the sport in a wheelchair [3–5]. The differentiation between the two classes of wheelchair users lies in their abilities. The abilities of athletes according to their classification were studied through temporal and technical aspects in two different studies, which are two important characteristics of athletes' performance [10,11]. These two studies analyzed technical and temporal variables via video analysis of matches [10,11]. A total of 25 matches of men's singles events performed during the 2018 Brazilian Para-badminton Championship were filmed (WH1: *n* = 10; WH2: *n* = 15) for the study of Mota Roibeiro and de Almada [11], and 20 matches from the men's singles category at the 11th World PBd Championship held in Ulsan, South Korea, in 2017 were analyzed for the study of Strapasson [10].

### 3.1. Temporal Effect

The following temporal parameters were measured in both studies [10,11]:

- Total playing time: time between the first game service to the last point of the game;
- Rally time: time between contact with shuttle during service and end of point;
- Effective time: time accounted for the sum of rallies;
- Working density: ratio between the effective time and the rest time;
- Work load: ratio between the total playing time and the effective time and displays the working relationship during the game. The lower the value, the greater the intensity of the game.

The comparison of these temporal parameters between the classes, depending on the study under consideration, is presented in Table 2.

From a temporal standpoint, the studies demonstrate that matches involving WH2 athletes exhibit greater intensity than those involving WH1 athletes [10,11]. As proposed by Strapasson et al. [10], this may be associated with a reduced number of strokes executed at a faster pace, decreased rally duration, and increased time required to retrieve grounded shuttlecocks, consequently extending the pause time for WH1 athletes. Thus, limitations in trunk mobility in WH1 athletes appear to directly impact the dynamics of badminton matches.

**Table 2.** Results of the comparison of temporal parameters between WH1 and WH2 depending on the study.

| | Strapasson (2021) [10] | p | ES | Mota Ribeiro and Almeida (2020) [11] | p | ES |
|---|---|---|---|---|---|---|
| Total playing time [s] | / | / | / | WH2 > WH1 | 0.037 | 0.44 |
| Rally time [s] | WH2 > WH1 | 0.05 | 0.2 | NS | 0.073 | 0.38 |
| Effective time [s] | / | / | / | WH2 > WH1 | 0.009 | 0.53 |
| Rest time [s] | WH2 < WH1 | <0.001 | 0.1 | WH2 < WH1 | 0.026 | 0.23 |
| Working density | / | / | / | WH2 > WH1 | 0.009 | 0.48 |
| Work load | / | / | / | WH2 > WH1 | 0.030 | 0.56 |

WH2 > WH1: the measured parameter is significantly higher for class WH2 compared to class WH1; WH2 < WH1: the measured parameter is significantly higher for class WH1 compared to class WH2; NS: non-significant; /: not measured by the study; p: p-value; ES: effect size.

### 3.2. Technical Effect

The studies measured several technical parameters defined in Table 3.

**Table 3.** Definition of the various technical parameters measured.

| Technical Parameters | Definitions |
|---|---|
| Total stroke (TSt) | Total number of strokes performed during a match |
| Frequency of strokes (FS) [stroke/s] | Number of strokes performed per second |
| Clear (C) | Stroke played in the mid-court over the net, with a high, deep trajectory for play at the opponent's end of the court |
| Drive (D) | Tense, dynamic stroke played in the mid-court over the net with a horizontal trajectory for play at the opponent's middle of the court |
| Drop shot (DS) | Curved trajectory stroke for play next to the net at the front of the court of your opposing player |
| Lob (L) | Stroke played close to the net to send the shuttlecock over the opponent's head and into the back of the court, giving the player time to return to the mid-court |
| Net shot (NS) | Stroke played close to the net and under the net for play next to the net at the front of the court of your opposing player |
| Smash (S) | Performed at the back of the court, it is a powerful downward stroke, hit flat over the head to try and score the point |
| Block (B) | Stroke struck from mid-court in response to a fast shot, removing almost all the speed of the shuttlecock and landing in the opponent's forward zone |
| Net lift (NL) | Stroke made at the net and going to the back of the court, requiring a certain height to pass clearly over the opponent |
| Short forehand service (SFS) | Short serve with your racket while the back of your hand is facing the shuttle |
| Long forehand service (LFS) | Long serve with your racket while the back of your hand is facing the shuttle |
| Short backhand service (SBS) | Short serve with your racket while the front of your hand is facing the shuttle |
| Long backhand service (LBS) | Long serve with your racket while the front of your hand is facing the shuttle |
| Total service (TSe) | Total number of services performed during a match |
| Winning points (WP) | Total number of points won by an attack by the athlete who provoked the opponent's error under pressure |
| Unforced error (UE) | Total number of out and net errors when the player is not under excessive pressure from the opponent and has the possibility of effective strokes to win the point |

In wheelchair badminton, strokes such as the drive and smash are considered aggressive, attacking strokes. Conversely, strokes such as block and clear are considered defensive. The comparison of technical parameters between the classes, depending on the study under consideration, is presented in Table 4.

**Table 4.** Results of the comparison of the number of strokes per technical parameter between WH1 and WH2 depending on the study.

| | Strapasson (2021) [10] | *p* | ES | Mota Ribeiro and de Almeida (2020) [11] | *p* | ES |
|---|---|---|---|---|---|---|
| TSt | WH2 > WH1 | <0.001 | 0.4 | WH2 > WH1 | 0.007 | 0.54 |
| FS [stroke/s] | / | | | NS | 0.705 | 0.08 |
| C | WH2 > WH1 | <0.001 | 0.3 | NS | 0.112 | 0.34 |
| D | WH2 > WH1 | <0.001 | 1.3 | WH2 > WH1 | 0.053 | 0.40 |
| DS | WH2 > WH1 | <0.001 | 0.4 | NS | 0.155 | 0.30 |
| L | WH2 > WH1 | <0.001 | 0.5 | / | / | / |
| NS | WH2 > WH1 | <0.001 | 0.5 | WH2 > WH1 | <0.001 | 0.68 |
| S | WH2 > WH1 | <0.001 | 0.8 | WH2 > WH1 | <0.001 | 0.77 |
| B | / | / | / | WH2 > WH1 | <0.001 | 0.67 |
| NL | / | / | / | NS | 0.077 | 0.38 |
| SFS | WH2 > WH1 | <0.001 | 0.5 | NS | 0.958 | 0.01 |
| LFS | WH1 > WH2 | <0.001 | 0.6 | NS | 0.388 | 0.20 |
| SBS | WH2 > WH1 | <0.001 | 0.5 | NS | 0.876 | 0.03 |
| LBS | WH1 > WH2 | 0.037 | 0.3 | NS | 0.141 | 0.32 |
| TSe | / | / | / | WH2 > WH1 | 0.009 | 0.53 |
| WP | / | / | / | NS | 0.933 | 0.02 |
| UE | / | / | / | WH2 > WH1 | <0.001 | 0.64 |

WH2 > WH1: the measured parameter is significantly higher for class WH2 compared to class WH1; WH1 > WH2: the measured parameter is significantly higher for class WH1 compared to class WH2; NS: non-significant; /: not measured by the study; *p*: *p*-value; ES: effect size.

Among the large number of different technical parameters measured, it seems that WH2 athletes employed more aggressive strokes (net shot, drive, and smash) in contrast to the WH1 athletes and executed more total strokes than WH1 players [10,11]. Given that WH2 athletes have more effective time at their disposal than WH1 athletes, we can assume that this accounts for their greater number of total strokes. As for drives, smashes, and net shots, they necessitate either significant forward flexion of the trunk or backward extension. It is reasonable to infer that the trunk instability of WH1 athletes restricts their proficiency in executing strokes that demand greater postural control.

However, the variability in results pertaining to technical parameters across various studies is evident despite the alignment in the parameters under investigation. These discrepancies may be attributed to differences in the number of matches analyzed. For instance, Strapasson [10] scrutinized 20 matches, whereas Mota Ribeiro and Almeida examined 25. Furthermore, the heterogeneity of player skill levels, spanning from international to national standards, across different studies could potentially contribute to the observed disparities. Given these substantial variations, establishing definitive technical characteristics specific to athlete classifications is beset with inherent complexities. Due to the disparities in results, elucidating the differences between WH1 and WH2 athletes is challenging.

### 3.3. Practical Implications

The results from the previous sections suggest that athletes in the WH2 category engage in more intense matches than those in the WH1 category. This observation should be taken into consideration, especially in the planning of training sessions. It might be prudent to propose less intensive sessions for WH1 athletes compared to WH2 athletes to prevent early fatigue. Longer or more regular rest periods could also be considered.

Regarding the higher number of aggressive shots executed by WH2 athletes compared to WH1 athletes, strategies for court positioning could be contemplated, taking into account the technical specificities of the athletes, especially in doubles. Indeed, adapting the athletes' positions on the court to favor attack by WH2 and defense by WH1 would be interesting. Additionally, game strategies could be devised based on the principle of attack by WH2, and defense by WH1 must be taken into consideration during athletes' training to offer

them something that aligns perfectly with their needs. Given these specificities, WH1 athletes should train for shorter periods and less intensively than WH2 athletes to avoid athlete fatigue due to overtraining.

### 3.4. Studies Analysis

To better understand the results presented in this section, it is also crucial to gain a better understanding of the study protocols. Indeed, both cited studies rely on a video analysis technique with variable criteria [10,11]. Firstly, the number of analyzed matches appears to have been chosen randomly in both studies. While in the study by Mota Ribeiro and de Almeida [11], the number of matches analyzed per class is specified, Strapasson's [10] study does not mention it, potentially indicating a disparity in the observed matches for one of the two categories and influencing the obtained results. Additionally, the process of inclusion and exclusion of analyzed matches is detailed in Mota Ribeiro and Almeida's [11] study but absent in Strapasson's [10] study, which may suggest differences in the analyzed matches and influence the results obtained. Finally, although reliability tests were conducted in both studies to validate the match analysis by a single examiner, control by a second examiner could have strengthened the results obtained in both studies [10,11]. These particularities in the study protocols must be taken into account when interpreting the results. Studies based on the measurement of biomechanical variables could be beneficial both for standardizing test protocols and for investigating data that could enhance the understanding of wheelchair badminton.

## 4. Propulsion Technique

The practice of wheelchair badminton and its internal logic imposes movements on the players: the athletes successively repeat forward and backward propulsion. Three studies have looked at backward propulsion with a view to preventing the risk of injury [12–14]. Indeed, since forward propulsion is predominantly used by wheelchair users, it leads to overuse of the upper limbs, which can accentuate the risk of secondary pathologies such as tendonitis or rotator cuff syndromes [12–14]. Backward propulsion has been designed as a protective solution to the overuse of forward propulsion. In the context of wheelchair badminton, backward propulsion is mandatory and frequently used. Its study is therefore a necessity, both for injury prevention and athlete performance. Three studies aimed to compare the physiological and biomechanical effects induced by two directions of propulsion on able-bodied subjects [12–14]. Each participant had to propel for 3 min at different rolling resistances in two studies [12,13] or different sub-maximal speeds in another [14] using a roller ergometer to measure different kinetic and kinematics parameters. Linden et al. [12] utilized a setup with independent wheels and a backless bench, while Salvi et al. [13] used an everyday wheelchair, and Mason et al. [14] employed a sports wheelchair. Mason et al. [14] also incorporated the use of an instrumented wheel. In addition to these measurement tools, physiological data were collected through indirect calorimetry for Linden et al. [12], open circuit spirometry for Salvi et al. [13], and a breath-to-breath system for Mason et al. [14].

### 4.1. Physiological Effect

The studies by Linden et al. [12] and Salvi et al. [13] measured their variables at different imposed rolling resistances, while Mason et al. [14] measured their variables at different imposed speeds. To facilitate understanding of the results presented, we will refer to an overall significant effect. The comparative analysis of physiological parameters obtained for forward and backward propulsion is presented in Table 5.

The findings of studies conducted by Mason et al. [14] and Salvi et al. [13] indicate that backward propulsion induces higher cardiorespiratory responses, overall intensity of effort, physiological stress, metabolic load, and rating of perceived exertion at high imposed rolling resistance or speeds [13,14]. Indeed, oxygen uptake and heart rate increase in backward propulsion [13,14], as well as the pulmonary ventilation, blood lactate, and rating

of perceived exertion [13]. As Mason et al. [14] noted, these findings may be attributed to an ergonomic wheelchair configuration not suited for backward propulsion. In fact, wheelchair seats are typically positioned to optimize forward propulsion, which could potentially increase the physiological demands on the athlete in the opposite propulsion direction.

**Table 5.** Results of the comparison of physiological parameters between forward propulsion and backward propulsion depending on the study.

|  | Linden et al. (1993) [12] | Salvi et al. (1998) [13] | Mason et al. (2015) [14] |
|---|---|---|---|
| Oxygen uptake [L/min$^1$ or mL/kg/min$^2$] | FP > BP | BP > FP | BP > FP |
| Respiratory exchange ratio | NS | / | / |
| Pulmonary ventilation [L/min] | FP > BP | BP > FP | / |
| Heart rate [beats/min] | / | BP > FP | BP > FP |
| Blood lactate [mmol/L] | / | BP > FP | / |
| Rating of perceived exertion [points] | / | BP > FP | / |

FP > BP: the measured parameter is significantly higher for the forward propulsion compared to the backward propulsion; BP > FP: the measured parameter is significantly higher for the backward propulsion compared to the forward propulsion; NS: non-significant; /: not measured by the study.

However, those results are at odds with the findings of Linden et al. [12]. Indeed, Linden et al. [12] note an increase in oxygen uptake and pulmonary ventilation in forward propulsion compared to backward propulsion, while Salvi et al. [13] and Mason et al. [14] show the opposite. The protocol employed in Linden et al.'s [12] study may account for these disparities in outcomes. Indeed, Linden et al. [12] utilized a setup where a backless bench was placed between two independently moving wheelchair wheels on an ergometer to simulate wheelchair propulsion instead of using an actual wheelchair. This configuration failed to accurately replicate the characteristics of wheelchair propulsion. Notably, the device used by Linden et al. [12] lacked a backrest. The absence of a backrest could have encouraged the use of back extensors, which are beneficial during backward propulsion but may have negatively impacted forward propulsion, especially at high rolling resistances or speeds [13,14]. An experiment so far from a real wheelchair propulsion condition can have an impact on the results obtained.

### 4.2. Biomechanic Effect

Three studies were conducted to examine the effects of propulsion direction on kinematics and propulsion technique with several different parameters measured [12–14]. As previously mentioned, the studies measured their variables at different imposed rolling resistance [12,13] or speeds [14]. To facilitate understanding of the results, we will refer to an overall significant effect. The comparative analysis of propulsion technique parameters obtained for forward and backward propulsion is presented in Table 6.

The results of these studies show that backward propulsion leads to challenges in applying sufficient forces on the handrim, necessitating adjustments in the propulsion technique's kinematics, especially at high rolling resistances or speeds [12–14]. Indeed, the findings of the study conducted by Mason et al. [14] reveal an increase in the forces (peak and mean resultant forces, mean tangential force, peak and mean radial forces, vertically downward maximal force, and mean lateral force) generated at the handrim during forward propulsion compared to backward propulsion, along with an increase in the rate of force development. It is noteworthy that the minimum vertical downward force is the only force to exhibit a higher value in backward propulsion than in forward propulsion, although it does not contribute significantly to wheelchair propulsion. Additionally, the studies identified a substantial increase in push time [14] and a decrease in strike rate and push frequency in backward propulsion compared to forward propulsion, indicating a significant alteration in propulsion technique in this configuration [12,13]. These changes could lead to reduced performance in a sporting context. Indeed, it could be difficult to reach maximum velocities if the forces required to propel the wheelchair are not sufficiently applied.

**Table 6.** Results of the comparison of force and technical propulsion parameters between forward propulsion and backward propulsion depending on the study.

| | Linden et al. (1993) [12] | Salvi et al. (1998) [13] | Mason et al. (2015) [14] |
|---|---|---|---|
| Work [J] | / | / | NS |
| Peak and mean resultant forces [N] | / | / | FP > BP |
| Mean tangential forces [N] | / | / | FP > BP |
| Peak and mean radial force [N] | / | / | FP > BP |
| Vertically downward maximal force [N] | / | / | FP > BP |
| Vertically downward minimal force [N] | / | / | BP > FP |
| Mean lateral force [N] | / | / | FP > BP |
| Fraction of effective force [%] | / | / | BP > FP |
| Rate of force development | / | / | FP > BP |
| Push frequency [push/s] or strike per minute [strike/min] | FP > BP | / | NS |
| Push angle [°] | / | / | FP > BP |
| Push time [s] | / | / | BP > FP |
| Strike rate [push] | / | FP > BP | / |
| Mechanical efficiency | BP > FP | / | / |
| Revolution per minute | NS | / | / |

FP > BP: the measured parameter is significantly higher for the forward propulsion compared to the backward propulsion; BP > FP: the measured parameter is significantly higher for the backward propulsion compared to the forward propulsion; NS: non-significant; /: not measured by the study.

However, it is important to note that Linden et al. [12] observed an increase in mechanical efficiency in backward propulsion compared to forward propulsion, indicating an increase in propulsion efficiency. As mentioned in the preceding section, this result should be interpreted cautiously due to the experimental protocol significantly deviating from ecological conditions, posing a potential impact on the obtained outcomes. Also noteworthy is that Mason et al. [14] found a higher effective force fraction in backward propulsion compared to forward propulsion, contrary to their previous findings. The authors explain that this increase in the fraction of effective force may result from a modification in the "grasping" technique, characterized by a slower and more extended approach, reinforced by an increase in vertical downward force [14]. This suggests that less force was wasted during backward propulsion, although it does not necessarily imply greater efficiency compared to forward propulsion [14].

### 4.3. Practical Implications

The initial findings in this section indicate that backward propulsion induces higher cardiorespiratory responses, overall intensity of effort, physiological stress, metabolic load, and rating of perceived exertion, particularly at high imposed rolling resistance or speeds. Considering these results, incorporating targeted muscle strengthening for the primary muscle groups involved in backward propulsion phases could potentially mitigate the physiological negative effects associated with this direction of propulsion.

The subsequent findings from these studies reveal that backward propulsion poses challenges in generating sufficient forces on the handrim, requiring adjustments in the kinematics of propulsion techniques, particularly at high rolling resistances or speeds [12–14]. This emphasizes the significance of adapting wheelchair ergonomics, specifically the seat, to accommodate both forward and backward propulsion. A more rear-facing seat could potentially minimize alterations in propulsion technique during backward propulsion. Additionally, identifying individual challenges related to backward propulsion could pave the way for strategically adjusting the athlete's positioning on the field to effectively address these issues. Indeed, a slightly more rear-centered court positioning could be recommended to proactively address challenges associated with backward propulsion.

*4.4. Studies Analysis*

To better understand the results of the three studies, it is important to better understand the protocol of these studies [12–14]. The protocol of Linden et al. [12] significantly differs from ecological conditions and the studies conducted by Salvi et al. [13] and Mason et al. [14], as discussed in the previous sections. Indeed, Linden et al. [12] utilized a setup where a backless bench was placed between two independent wheelchair wheels on an ergometer to simulate wheelchair propulsion instead of using an actual wheelchair. This configuration failed to accurately replicate the characteristics of wheelchair propulsion [12]. It is also relevant to examine the protocols of the studies by Mason et al. [14] and Salvi et al. [13]. Indeed, these two studies were conducted on a population of able-bodied subjects to avoid any inherent learning bias related to forward propulsion. Given that the level of impairment influences athletes' performance, the results of these studies are not entirely generalizable to a population of athletes with motor disabilities [13,14]. Additionally, the use of the roller ergometer in both studies neutralizes the rolling resistance of the front wheels and may potentially underestimate certain variables such as power or oxygen consumption [9,12–14]. On the other hand, Mason et al. [14] coupled the use of a roller ergometer with a Smartwheel, increasing the total weight of the wheelchair, which can also modify the rolling resistance of the wheelchair and impact the obtained results. Conducting studies on a population of wheelchair badminton players in ecological conditions would be interesting to better understand the impact of propulsion directions on the performance of these athletes.

**5. Use of the Racket**

The use of the racket is a particularity of wheelchair badminton, which athletes cannot abstain from. Therefore, it is interesting to evaluate the impact of this badminton racket on their propulsion. Although it is not possible to eliminate the BR during propulsion, potential solutions include modifying the ergonomics of the wheelchair handrim or adjusting the grip of the racket and handrim. Additionally, adaptations in athletes' physical preparation could be implemented to account for the impact of the racket. Thus, a better understanding of this tool and its implications could lead to beneficial modifications in athletes' performance and contribute to reducing the risk of injuries. One study focused on examining the influence of employing a badminton racket on the propulsion's kinematics [15], and one short communication focused on the impact of the racket on muscular parameters during wheelchair propulsion [16]. In the study by Alberca et al. [15], 16 novice able-bodied subjects who underwent wheelchair badminton training performed a test at a stabilized submaximal speed of 5 km/h. They were required to propel the wheelchair at this stabilized speed along a straight line of 20 m with and without a racket. Only one sports wheelchair was used for the tests, which was equipped with two instrumented wheels to measure kinematics and kinetics parameters [15]. In the short communication by de Fukui et al. [16], six healthy men performed maximal repetitions of 30 cm forward–back sprints using a wheelchair for 20 s under two conditions: with and without a racquet. The number of sprints, muscle hardness of the ulnar carpi flexor using a muscle hardness tester, and deoxygenated hemoglobin (HHb) using near-infrared spectroscopy (NIRS) were measured before and after each condition.

*5.1. Kinematic Effect*

One study focused on examining the influence of employing a badminton racket on the propulsion's kinematics [15]. Their results concerning the kinetic parameters of propulsion reveal that the use of the racket alters the force application on the handrim in a manner that increases generated forces while decreasing the efficiency of propulsion [15]. In fact, their study revealed that the rate of rise, maximum power output, and push angle are higher during the racket condition compared to the without racket condition [15]. Furthermore, the maximal propulsive moment increases in racket conditions, which is associated with an elevation in maximal total force and a reduction in the fraction of

effective force. This suggests that while the propulsive moment increases, non-contributory forces to the propulsion, such as radial force, increase equivalently or even more, resulting in a decrease in the fraction of effective force and, consequently, a reduction in the efficiency of athlete propulsion. According to Alberca et al. [15], these kinetic findings can be explained by the coupling between the hand holding the racket and the handrim. The challenge of gripping on the handrim with the presence of the racket compels athletes to adapt their propulsion kinetics. This adaptation involves increasing the forces and power exerted on the handrim to sustain a consistent velocity. This could lead to a reduction in performance as well as an increase in injury risks.

### 5.2. Temporal Effect

The same study focused on examining the influence of using a badminton racket on the temporal parameters of wheelchair propulsion [15]. Their temporal findings suggest an alteration in the propulsion technique when employing a racket [15]. Specifically, it seems that the motion becomes shorter and quicker with the use of a racket because push time, cycle time, and push angle decrease without a racket compared to with a racket [15]. In their study, Alberca et al. [15] suggest that these temporal results can be attributed to challenges in coordinating the hand holding the racket with the handrim, as well as the weight of the racket. Indeed, the difficulties in grasping the handrim with the racket, coupled with its weight, may lead athletes to shorten their propulsion gesture, resulting in an increase in their speed of movement. This adjustment could result in a diminished and less efficient force application to the handrim, potentially compromising the athlete's performance.

### 5.3. Muscular Effect

Only a short communication addressed the impact of the racket on muscular parameters during wheelchair propulsion [16]. Their findings highlight that the muscle hardness of the ulnar carpi flexor and deoxygenated hemoglobin are greater with the racket compared to the condition without the racket. They also added that the number of sprints performed in propulsion with a racket was reduced compared with propulsion without a racket. Since this is a short communication, explaining the results is challenging. Nevertheless, these initial results seem to indicate that propulsion with a racket increases the load on the upper limb holding the racket and increases athlete fatigue.

### 5.4. Practical Implications

Initial results concerning the kinetic parameters of propulsion reveal that the use of the racket alters the force application on the handrim in a manner that increases generated forces while decreasing the efficiency of propulsion [15]. Exploring the development of a new handrim for athletes, with the aim of enhancing friction between the racket handle and the handrim, could be intriguing to reduce coupling challenges between the two. Indeed, two studies in wheelchair tennis have investigated testing new handrim designs to improve the grip between the hand with the racket and the handrim [18,19]. Rietveld et al. [19] have shown a reduction in negative power and higher mechanical efficiency with their new handrim design. Similar research could also be conducted in the field of wheelchair badminton. However, implementing such solutions requires some time. As a more immediate measure, it might be conceivable to modify the handrim covering to increase the friction effect.

Secondly, temporal findings suggest a modification in the propulsion technique when using a racket: athletes push shorter distances and more quickly with the use of a racket [15]. Proposing training with feedback could optimize the backward propulsion gesture. It is well-established that the semi-circular propulsion pattern offers the best performance for a wheelchair athlete [20–22]. To guide the athlete toward this pattern of propulsion to extend their push time and frequency using haptic feedback propulsion simulator systems, as suggested by Blouin et al., could be considered.

Finally, the results regarding the potential effects of using the racket on the muscles involved in propulsion do not allow for the establishment of recommendations, as they stem from the findings of a short communication.

*5.5. Studies Analysis*

Upon analyzing the protocols of the studies mentioned in the preceding sections, it is important to note that these investigations involved novice able-bodied participants who received wheelchair badminton training in the case of Alberca et al. [15] and were entirely novice participants for Fukui et al. [16]. As mentioned earlier, although studies on able-bodied individuals aim to minimize learning biases, their results may not be entirely applicable to wheelchair athletes [23–25]. Furthermore, Alberca et al.'s [15] study, despite being conducted on the field, only focused on submaximal exercise and forward propulsion. However, the internal dynamics of wheelchair badminton involve both maximal forward and backward propulsion. Variances in results may arise when comparing the study's outcomes with those of maximal forward and backward propulsion. Lastly, Fukui et al.'s [16] study employed a two-way repeated-measures ANOVA for result analysis. Given the small study population (six participants), it is possible that the analysis may have overestimated the significance of the measured variables, influencing the obtained results. Considering these insights, investigating the use of the badminton racket among a population of wheelchair badminton athletes in both forward and backward propulsion would be pertinent.

## 6. Discussion

The aim of this narrative review was to provide a biomechanical overview of the literature on wheelchair badminton and its specificities: the athletes' abilities related to their classification, the direction of propulsion, and the use of a racket. The history of wheelchair badminton shows that this sport is relatively young, which is confirmed by the lack of scientific studies in this field [10–16]. Given the lack of biomechanical data for wheelchair badminton, technical, temporal, and physiological data had to be included in this narrative review. Indeed, this review relates in an almost exhaustive way to the articles on the topic of wheelchair badminton.

These various studies represent valuable contributions to sports practitioners and contribute to a better understanding of this discipline [10–16]. Indeed, these studies reveal that WH2 athletes engage in more intense matches and execute more offensive shots compared to WH1 athletes [10,11]. Additionally, backward propulsion induces higher cardiorespiratory responses, overall intensity of effort, physiological stress, metabolic load, and perceived exertion ratings, along with challenges in generating sufficient forces on the handrim. This necessitates adjustments in the kinematics of propulsion techniques, especially at high rolling resistances or speeds, compared to forward propulsion, potentially resulting in performance decreases [12–14]. Lastly, the use of a badminton racket alters the force application on the handrim, increasing generated forces while decreasing propulsion efficiency and modifying the propulsion technique with shorter and quicker pushes, which could negatively impact performance [15]. More concretely, the results of this narrative review indicate the following points for armchair badminton coaches and/or athletes:

- To reduce the intensity of training sessions for WH1 athletes compared to WH2 athletes to prevent early fatigue;
- To adapt the athletes' positions on the court to favor attack by WH2 and defense by WH1;
- To incorporate targeted muscle strengthening for the primary muscle groups involved in backward propulsion;
- To rear-center court position to proactively address challenges associated with backward propulsion;
- To explore the development of a new handrim for athletes to improve grip between the handrim and the hand handle of the badminton racket;

- To modify the handrim covering to increase the friction effect.

Figure 1 summarizes the main findings of this study.

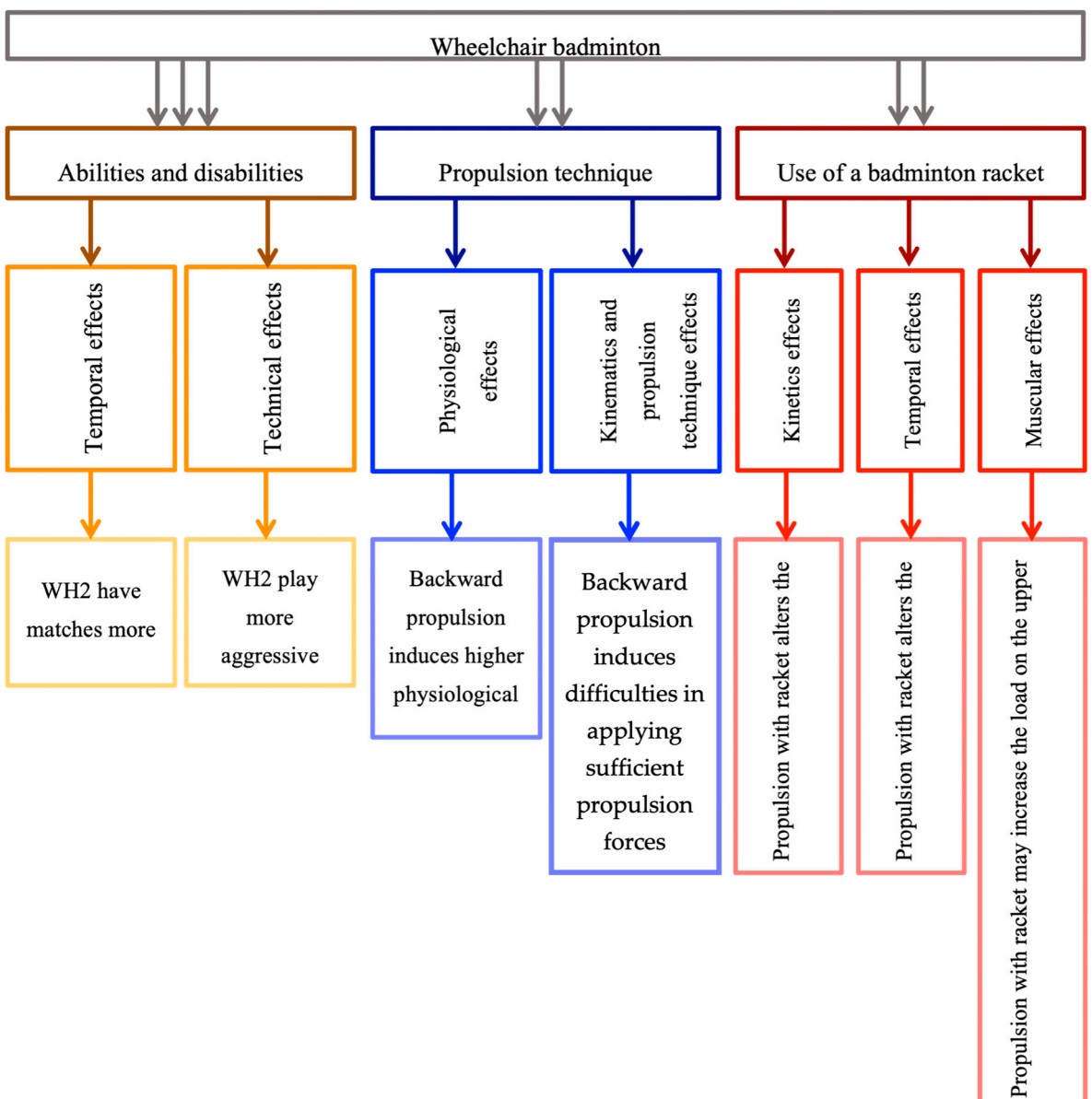

**Figure 1.** Main findings of the narrative review.

However, it is important to be cautious with the results obtained for several reasons. Indeed, firstly, no biomechanical characteristics are known for the two classes of wheelchair badminton. The absence of biomechanical data presents a significant constraint in comprehending a sporting discipline. Previous studies in wheelchair sports, such as wheelchair rugby, wheelchair tennis, or wheelchair basketball, have undergone thorough biomechanical analyses [26–31], contributing substantially to an enhanced understanding of the kinetics and kinematics of propulsion—pivotal biomechanical dimensions essential for evaluating athletes' performance and identifying injury risks. Moreover, Mason et al. [14] authored a comprehensive review consolidating ergonomic considerations for wheelchairs conducive to optimal performance in wheelchair court sports, drawing from a multitude of studies focused on wheelchair biomechanics [17,26,27,29–32]. Consequently, the dearth of biomechanical data could impede a comprehensive understanding of wheelchair bad-

minton, hindering efforts to enhance athletes' performance, optimize propulsion, and mitigate injury risks.

Secondly, studies concerning propulsion technique and racket use have been carried out in the laboratory. The issue of laboratory testing has been extensively explored in the literature. Due to constraints associated with conducting laboratory tests (availability, test duration, etc.), field tests have become a preferable solution for wheelchair athletes and coaches [33]. They offer the advantage of rapidly testing many athletes simultaneously and assessing them under ecological conditions. The results from field tests could potentially be more relevant than those from laboratory tests. Therefore, it is essential that studies under ecological conditions be undertaken in the field of wheelchair badminton.

Finally, some studies have been conducted on novice able-bodied subjects. Considering that the level of impairment influences athletes' performance, the results of these studies are not entirely applicable to a population of athletes with motor disabilities. Indeed, as indicated by Vanlandewijck et al. [28], trunk movements are fundamental mechanisms for generating force in high-resistance propulsion. These movements impact the rolling resistance of the athlete's wheelchair. However, increased rolling resistance can result in reduced propulsion velocity, longer cycle and push times, increased mean power output, and propulsion forces [23–25]. Trunk mobility also influences the orientation of the push angle on the handrim, a parameter linked to propulsion efficiency [28]. Given these findings and the fact that some wheelchair badminton players have limited or no abdominal capabilities, it would be necessary to conduct studies on a population of wheelchair badminton players. So, while we can draw conclusions from the various studies presented in this review, we must remain cautious about drawing conclusions due to the diversity of protocols employed in these studies, the populations used, and the lack of biomechanical parameters [10–16]. Therefore, further research is necessary to explore other perspectives and expand the field of study in the wheelchair badminton domain on biomechanics data and wheelchair badminton players on ecological conditions.

Considering these results and discussions, several perspectives for research and development emerge. Indeed, comparing and characterizing biomechanically athletes based on their classification could allow for more precise adjustments to their training programs and match strategies. Beyond this aspect, a kinematic characterization of the different classes could give rise to an additional objective tool for the athlete classification process. Given that wheelchair badminton remains a relatively young sport, the limited development of its classification system underscores the potential importance of integrating an objective biomechanics on-field measure to determine athletes' classes, thereby constituting an asset. Regarding propulsion directions, an in-depth study with wheelchair badminton players could clarify the potentially negative impact of backward propulsion compared to forward propulsion, leading to ergonomic adaptations of the wheelchair to optimize performance in both propulsion directions. A modification to the wheelchair's backrest could enhance performance in backward propulsion while preserving advantages for forward propulsion. Finally, concerning racket usage, it would be conceivable to explore the effect on wheelchair badminton athletes and test modifications such as changes to the shape of the handgrip, the use of new materials for its coating, or the introduction of new grips on the racket handle to improve the grip between the hand and the handrim.

**Author Contributions:** Conceptualization, I.A., B.W., F.C. and A.F.; methodology, I.A.; software, I.A.; validation, A.F., B.W. and F.C.; formal analysis, I.A.; investigation, I.A. and A.F.; resources, I.A.; data curation, I.A. and F.B.; writing—original draft; preparation, I.A.; writing—review and editing, I.A., B.W., A.F., F.B. and F.C.; visualization, I.A.; supervision, A.F.; project administration, A.F.; funding acquisition, A.F. All authors have read and agreed to the published version of the manuscript.

**Funding:** This work was supported by a French government grant managed by the Agence Nationale de la Recherche (ANR) under the "France 2030" program, reference ANR-19-STHP-0005.

**Data Availability Statement:** Data is contained within the article.

**Acknowledgments:** The authors of this study certify that the results of the study are presented clearly, honestly, and without fabrication, falsification, or inappropriate data manipulation and state that the results of the present study do not constitute an endorsement by ACSM.

**Conflicts of Interest:** The authors declare no conflicts of interest. The funders had no role in the design of the study; in the collection, analyses, or interpretation of data; in the writing of the manuscript; or in the decision to publish the results. The authors alone are responsible for the written content of this article.

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
