# Peer review of "Wheelchair Badminton: A Narrative Review of Its Specificities"

_2673-7078, doi:10.3390/biomechanics4020012_

Round 1

Reviewer 1 Report

Comments and Suggestions for Authors

The generic scope of the paper entitled "Wheelchair badminton: a narrative review of its specificities", provides a comprehensive overview of wheelchair badminton and its specificities. The paper aims to cover various aspects of wheelchair badminton, including its origin and growth, the classification system of athletes into WH1 and WH2 classes based on abilities and disabilities, the temporal and technical effects on game dynamics, an exploration of propulsion techniques, and an analysis of the impact of racket usage. The review incorporates existing studies in the field to contribute valuable insights for sports practitioners and aims to enhance the overall understanding of wheelchair badminton as a sport. The authors also acknowledge the relative youth of wheelchair badminton and the scarcity of scientific studies in this domain, highlighting the need for further research to expand the knowledge in this area.

1. GENERIC COMMENTS

1.1       Paper Strengths:

·       Clear Introduction: The introduction provides a clear overview of wheelchair badminton, its origin, and its growth, establishing the context for the review.

·       Comprehensive Material and Methods Section: The materials and methods section clearly outlines the approach taken for the narrative review, justifying the choice over a systematic review due to the novelty and limited studies in the field.

·       Detailed Classification System: The paper delves into the wheelchair badminton classification system, providing a detailed explanation of the two classes (WH1 and WH2) based on abilities and disabilities. This adds depth to understanding the athlete's categorization.

·       In-Depth Analysis of Temporal and Technical Effects: The discussion on temporal and technical effects, especially in Sections 3.1 and 3.2, provides a thorough analysis of how different aspects affect the game intensity, strokes, and strategies of WH1 and WH2 athletes.

·       Detailed Exploration of Propulsion Technique: The review extensively explores the physiological and biomechanical effects of propulsion techniques, shedding light on the challenges faced by athletes and the potential impact on performance.

·       Insightful Analysis of Racket Usage: The section on the use of the racket discusses kinetic, temporal, and muscular effects, offering valuable insights into how the equipment affects propulsion kinetics and athlete performance.

·       Comprehensive Overview: The paper provides a comprehensive overview of wheelchair badminton, covering various aspects such as abilities and disabilities, propulsion technique, and the use of a racket.

·       Inclusion of Studies: The incorporation of several studies adds depth to the review, giving readers insights into the existing research in the field.

·       Clear Structure: The paper has a clear structure with distinct sections for introduction, materials and methods, and discussions on different aspects of wheelchair badminton.

 1.2           Paper Weaknesses:

·       Limited Research Population: Some sections mention studies conducted on novice able-bodied participants. It's crucial to emphasize the need for studies specifically conducted on wheelchair badminton athletes to ensure the relevance of the findings.

·       Lack of Biomechanical Characteristics: The paper notes the absence of biomechanical characteristics for the two classes of wheelchair badminton athletes. This is a significant knowledge gap that should be addressed in future research.

·       Need for Ecological Context in Studies: The discussion on the effects of backward propulsion highlights the importance of studies in a sporting and ecological context. The paper rightly identifies the need for further investigations but could stress this point more strongly.

·       Caution in Interpreting Findings: The paper acknowledges discrepancies in results and highlights potential issues with certain study protocols. It's crucial to stress caution in interpreting findings and the need for standardized methodologies in future research.

·       Lack of Critical Analysis: While the paper presents findings from various studies, it could benefit from a more critical analysis of the methodologies employed in these studies. Discussing the limitations of each study and addressing potential biases would enhance the paper's credibility.

·       Limited Research on Wheelchair Badminton Athletes: The absence of biomechanical characteristics for the two classes of wheelchair badminton and the reliance on studies with able-bodied subjects raises questions about the applicability of the findings to wheelchair athletes. The paper should acknowledge this limitation.

·       Need for Further Research: The conclusion emphasizes the need for further research, which is a valid point. However, the paper might benefit from offering specific recommendations for future studies, such as exploring the biomechanical characteristics of wheelchair badminton athletes or conducting field studies with experienced wheelchair athletes.

Recommendation: Overall, the paper provides a comprehensive review of wheelchair badminton, covering various aspects. However, it should be revised to emphasize the necessity of studies with wheelchair badminton athletes, address the lack of biomechanical characteristics, and highlight the importance of ecological contexts in research. The caution in interpreting findings due to variations in methodologies should be stressed. With these revisions, the paper could be considered for acceptance.

2. SPECIFIC COMMENTS

2.1           Introduction

2.1.1. Strengths:

·       Clarity and Conciseness: The introduction is clear and concise, providing a brief yet comprehensive overview of the global popularity of badminton and the emergence of wheelchair badminton as an adapted form.

·       Historical Context: The introduction effectively incorporates the historical context of wheelchair badminton, mentioning its origins in the 1990s and the initiative taken by German athletes to adapt the rules.

·       Recognition of Inclusivity: The introduction emphasizes the inclusivity of wheelchair badminton, both in terms of gender and disability, highlighting its alignment with rules established by the Badminton World Federation (BWF).

·       Global Growth and Achievements: The mention of substantial international growth and the inclusion of wheelchair badminton in the 2021 Tokyo Paralympic Games adds credibility and significance to the sport.

·       Identification of Specificities: The introduction successfully identifies specificities unique to wheelchair badminton, such as the classification system, distinct movements, and equipment similarities with able-bodied badminton.

·       Relevance of Racket Impact: By referencing studies in wheelchair tennis, the introduction establishes the relevance of investigating the impact of racket use on wheelchair badminton athletes, creating a strong rationale for the narrative review.

2.1.1. Weaknesses:

·       Potential Redundancy: The introduction briefly mentions the global popularity of badminton, which might be considered redundant, as the focus is primarily on wheelchair badminton. This part could be streamlined for conciseness.

·       Need for Citations: While the introduction refers to studies in wheelchair tennis showing the impact of a racket on an athlete's performance, it would strengthen the argument by providing specific citations for these studies.

·       Clarification on Classification System: The introduction introduces the wheelchair badminton classification system but could benefit from a bit more detail or a reference to where readers can find more information about these classes.

·       Inclusion of a Clear Thesis Statement: While the introduction outlines the specificities to be discussed in the narrative review, it could be strengthened by including a clear thesis statement that explicitly states the purpose and goals of the review.

Recommendation: The introduction is generally strong, providing a solid foundation for the narrative review. To enhance it further, consider streamlining the global popularity mentioned, providing citations for studies in wheelchair tennis, adding more details on the classification system, and incorporating a clear thesis statement.

2.2. Methods

2.2.1. Strengths:

·       Clear Research Scope: The Materials and Methods section clearly outlines the scope of the narrative review, identifying the specific features of wheelchair badminton to be examined, such as racket use, classification, and propulsion techniques.

·       Rationale for Narrative Review: The section provides a rationale for choosing a narrative review over a systematic review, considering the novelty of the sport and the limited scientific studies available. This decision is supported by a reference to a relevant narrative review by Bakatchina et al. [8].

·       In-Depth Exploration of Classification System: The detailed explanation of the wheelchair badminton classification system (WH1 and WH2 classes) adds depth to the understanding of how athletes are categorized based on their abilities and disabilities.

·       Structured Presentation of Temporal and Technical Effects: The division of the Abilities and Disabilities section into temporal and technical effects provides a structured and clear presentation of the research findings. The use of tables and subsections enhances readability.

·       Evidence-Based Analysis: The section incorporates findings from specific studies, referencing Strapasson et al. [13], Linden et al. [14], Salvi et al. [15], Mason et al. [16], and Alberca et al. [17], adding credibility to the narrative review.

·       Thorough Exploration of Propulsion Techniques: The section on propulsion techniques provides a comprehensive overview, considering physiological, biomechanical, and temporal effects, contributing to a holistic understanding of the impact of forward and backward propulsion in wheelchair badminton.

·       Detailed Examination of Racket Use: The section on the use of the racket explores various aspects, including kinematic, temporal, and muscular effects, contributing to a nuanced understanding of how the racket influences wheelchair badminton players.

2.2.1. Weaknesses:

·       Limited Information on Temporal and Technical Effects: While the section mentions specific temporal and technical parameters measured in studies, it lacks detailed explanations of these parameters and the implications of the findings.

·       Need for Additional Citations: Some statements mention studies without providing specific citations (e.g., "studies demonstrate," "studies show"). Including citations for such statements would enhance the transparency and credibility of the narrative review.

·       Absence of a Conclusion to the Materials and Methods Section: The section concludes abruptly without summarizing the methods employed for the review or outlining any limitations. A brief conclusion summarizing the methodological approach could enhance clarity.

Recommendations: To strengthen the section, consider including publication years for cited studies, providing more detailed explanations of temporal and technical parameters, adding citations for specific statements, and concluding the section with a summary of the methodological approach and potential limitations 

2.3           Discussion

2.3.1. Strengths:

·       Clear Aim and Overview: The Discussion section begins with a concise restatement of the review's aim, providing an overview of the scientific literature on wheelchair badminton and its specificities.

·       Acknowledgment of Young Nature of the Sport: The discussion acknowledges the relative youth of wheelchair badminton as a sport, which is supported by the limited scientific studies available in the field. This acknowledgment provides context for the scarcity of research findings.

·       Comprehensive Synthesis of Studies: The section effectively summarizes key findings from various studies, including insights into the differences between WH1 and WH2 athletes, the effects of backward propulsion, and the impact of using a badminton racket. The synthesis adds value to sports practitioners and contributes to a better understanding of the discipline.

·       Visual Aid for Summary: The inclusion of Figure 1 enhances the presentation by providing a visual summary of the main findings. This visual aid can facilitate quick comprehension for readers.

2.3.1. Weaknesses:

·       Limited Mention of Biomechanical Characteristics: The acknowledgment that "no biomechanical characteristics are known for the two classes of wheelchair badminton" could be considered a weakness. Biomechanical insights are crucial for a comprehensive understanding of the sport, and the absence of such information is a notable gap that could impact the discussion.

·       Concerns about Generalization: The section mentions that studies on propulsion technique and racket use were conducted on able-bodied subjects or novice able-bodied subjects. This raises concerns about the generalization of findings to wheelchair badminton athletes, and it underscores the need for dedicated studies involving wheelchair athletes.

·       Ambiguity in Phrasing: The phrase "backward propulsion induces an increase in physiological responses" could benefit from clarification. It would be helpful to specify which physiological responses are being referred to and how they impact performance.

Recommendations: To strengthen the discussion, consider addressing the limited information on biomechanical characteristics for wheelchair badminton athletes, providing clarity on the specific physiological responses affected by backward propulsion, and emphasizing the importance of future research involving wheelchair athletes in laboratory and field settings.

Comments on the Quality of English Language

The overall quality of the English language in the paper is strong. The language used is clear, concise, and generally well-structured. The document follows proper academic writing conventions and maintains a formal tone throughout.

Reviewer 2 Report

Comments and Suggestions for Authors

General comment:

The MS is easily readable and appears to cover what information is available on the physical aspect of WC badminton.

However, it is to ‘easy’ and much info that I would expect in an review is missing. There is no overview of the search and selection process, no overview of the literature found and there is no clear definition of what was and was not included in the review (psychological / social / mental aspects?). Lastly, the discussion does actually not contain a reflection of what these results mean for the sport: what info do we need, for what? The  fact that this is missing is a logical consequence of a lack of specificity in the research question.

The authors need to step up a bit ….

Line 18: define WH2 and WH1

L:ine 20: rather vague. Please make this sentence more informative. What responses? What decrements?

Line 22: could this be rephrased in a more informative format?

Line 50: superfluous reference. Please remove

Line 55: it might be narrative, but what sources were examined, what search strings and selection methods were used? Please add.

More in general, subjects like “temporal effect” and “technical effect” should be mentioned in the research question and in the methods. These were apparently search strings?. The same applies for propulsion technique.

Line 70: this should be mentioned under methods

Line 93: I would like to know more about these two papers: are their research methods comparable, what / how many subjects were measured, when and where were these observations made?

Line 108: this sounds like a circular argument, please comment

Line 116: which studies?

Line 162: this information should be given in the search results overview.

Line 170: how relevant is research on submaximal exercise wheelchair badminton?

Line 170: how were these measurements made? It appears that researchers used ergometers. Were measurements performed with or without the racket?

Line181: again I would like to see more information on the publications used. I would strongly suggest to add a table with an overview of search results.

Line 214: do not start a new section with a back referral

Line 228: again, details on the publications are missing.

Line 317: “temporals”?

Line 344; Discussion: Add a discussion on the relevance of the physiological measurements on ergometers for the interpretation of the intensity of WC badminton. Add a discussion on how values could be obtained and what information is really needed and for what.

At this point the MS is much on the level of “nice to know but so what”.

Round 2

Reviewer 1 Report

Comments and Suggestions for Authors

No comments

Comments on the Quality of English Language

No comments

Reviewer 2 Report

Comments and Suggestions for Authors

Thank you for the thorough and conscientious revision.